# Serum Lipoprotein Profiling by NMR Spectroscopy Reveals Alterations in HDL-1 and HDL-2 Apo-A2 Subfractions in Alzheimer’s Disease

**DOI:** 10.3390/ijms252111701

**Published:** 2024-10-31

**Authors:** Jonas Ellegaard Mortensen, Trygve Andreassen, Dorte Aalund Olsen, Karsten Vestergaard, Jonna Skov Madsen, Søren Risom Kristensen, Shona Pedersen

**Affiliations:** 1Department of Clinical Biochemistry, Aalborg University Hospital, 9000 Aalborg, Denmark; j.ellegaard@rn.dk (J.E.M.); srk@rn.dk (S.R.K.); 2Department of Biochemistry and Immunology, Lillebaelt Hospital, University Hospital of Southern Denmark, 7100 Vejle, Denmark; dorte.aalund.olsen@rsyd.dk (D.A.O.); jonna.skov.madsen@rsyd.dk (J.S.M.); 3Department of Circulation and Medical Imaging, Norwegian University of Science and Technology, 7491 Trondheim, Norway; trygve.andreassen@ntnu.no; 4Central Staff, St. Olavs Hospital HF, 7006 Trondheim, Norway; 5Department of Regional Health Research, Faculty of Health Sciences, University of Southern Denmark, 5230 Odense, Denmark; 6Department of Neurology, Aalborg University Hospital, 9000 Aalborg, Denmark; k.vestergaard@rn.dk; 7Department of Clinical Medicine, Aalborg University Hospital, 9000 Aalborg, Denmark; 8Department of Basic Medical Science, College of Medicine, Qatar University, QU Health, Doha 2713, Qatar

**Keywords:** Alzheimer’s disease, lipoproteins, blood, serum, nuclear magnetic resonance, biomarker

## Abstract

Identifying biomarkers for Alzheimer’s disease (AD) is crucial, due to its complex pathology, which involves dysfunction in lipid transport, contributing to neuroinflammation, synaptic loss, and impaired amyloid-β clearance. Nuclear magnetic resonance (NMR) is able to quantify and stratify lipoproteins. The study investigated lipoproteins in blood from AD patients, aiming to evaluate their diagnostic potential. Serum and plasma were collected from AD patients (*n* = 25) and healthy individuals (*n* = 25). We conducted a comprehensive lipoprotein profiling on serum samples using NMR spectroscopy, analysing 112 lipoprotein subfractions. In plasma, we measured unspecific markers of neuronal damage and AD hallmark proteins using single molecule array technology. Additionally, clinical data and cerebrospinal fluid biomarker levels were also collected to enrich our data. Our findings, after adjusting for age and sex differences, highlight significant alterations in two specific lipoproteins; high-density lipoprotein (HDL)-1 Apo-A2 (H1A2) and HDL-2 Apo-A2 (H2A2), both with area under the curve (AUC) values of 0.67, 95% confidence interval (CI) = 0.52–0.82). These results indicate that these lipoprotein subfractions may have potential as indicators of AD-related metabolic changes.

## 1. Introduction

Alzheimer’s disease (AD) stands as the foremost cause of dementia worldwide, contributing to a significant proportion of the global healthcare cost, mortality, and morbidity [1]. Therapeutic interventions have predominantly centred around the amyloid hypothesis [2], yet despite these efforts, clinical trials continue to encounter challenges and show inadequate outcomes [3]. Consequently, it has become imperative to explore new avenues and pathways to understand this disease better and effectively respond to the increasing demand to identify biomarkers associated with the disease [4,5]. Current diagnostics involve a battery of cognitive assessments, imaging techniques, and analysis of cerebrospinal fluid (CSF) biomarkers such as amyloid-β (Aβ) and tau isoforms. These methods enable the investigation of structural, functional, and molecular alterations associated with the disease [6,7]. In some instances, this diagnostic battery has been supplemented with measuring neurofilament light (Nf-L) levels in the CSF, which serve as a non-specific indicator for neuronal damage in neurodegenerative diseases [8]. Despite notable progress in terms of sensitivity, there are still several drawbacks that need to be addressed. These drawbacks encompass patient compliance issues regarding the collection of CSF through lumbar puncture and the availability of scanning equipment for general practitioners. For these diagnostic tools to be utilized effectively as screening measures, it is imperative to address and overcome these limitations [9,10]. Blood sampling is easily performed in relevant settings [11] and minimally invasive [12]. Furthermore, blood is in proximity to every organ, allowing organ markers to be detected, and therefore blood samples could be a valuable screening tool [12].

Recent insights into the role of lipid metabolism in AD pathogenesis highlight the potential of lipoproteins as biomarkers. Lipoproteins are vital in lipid transportation, constructed of a single outer phospholipid-cholesterol monolayer, with the hydrophilic end oriented outward. Their density and size stratify them in an opposite manner, where very low-density lipoproteins (VLDL) are the largest particles, with a density of 0.93–1.006 g/mL, and high-density lipoproteins (HDL) are the smallest particles, with a density of 1.063–1.21 g/mL. The embedded apolipoproteins in their membrane determine their functions [13,14]. Lipoproteins do not typically traverse the blood–brain barrier (BBB). However, cells within the brain can generate lipoproteins, primarily containing apolipoproteins (Apo) E and J. Lipoproteins in the circulatory system contain other apolipoproteins, such as ApoB-100 and ApoA-I [15]. As part of the pathogenesis of AD, the disruption of the BBB facilitates the identification of circulatory lipoproteins in CSF [16].

Additionally, studies in animal models have observed that the accumulation of lipid droplets in the AD brain occurs prior to amyloid aggregation, highlighting the potential significance of dysregulated lipoprotein metabolism in neurodegenerative disorders [17]. Furthermore, cholesterol and low-density lipoprotein (LDL) have been identified as risk factors for AD. In contrast, HDL has been proposed to have a protective role, as higher levels of this lipoprotein have resulted in better cognitive outcomes [18] and are inversely correlated with cerebral Aβ deposits [19]. However, studies have also indicated that the functionality of HDL may not solely rely on its quantity but rather its associated apolipoproteins [20].

In clinical practice, blood lipoprotein tests are routinely performed, typically using enzymatic reactions [21]. These assays only measure total cholesterol, HDL levels, and triglycerides, while LDL usually is estimated using calculations such as the Friedewald equation [22]. A conventional approach for comprehensive quantification of lipoproteins involves a time-consuming ultracentrifugation process, which physically separates lipoprotein fractions, including chylomicrons, VLDL, LDL, and HDL. Subsequently, these fractions undergo in-depth analysis [23]. A novel approach involves using commercially available proton nuclear magnetic resonance (^1^H NMR) spectroscopy to perform quantitative profiling of lipoproteins in serum aliquots [13]. In contrast to mass spectrometry, which some previous studies have used to investigate lipid changes in AD, NMR requires a non-destructive method for sample preparation, as well as providing high reproducibility [24].

Therefore, the main objective of this study is to explore the diagnostic potential of lipoprotein subfractions as blood-based biomarkers for AD, offering insights into metabolic changes and their correlation with clinical data. By identifying specific lipoprotein alterations in AD, we hope to contribute to the development of simpler, more effective diagnostic tools for this complex disease.

## 2. Results

### 2.1. Clinical Characteristics

The biochemical analyses included clinical test results, organ functionality, and neurodegeneration markers, all as previously reported [25]. Briefly, the biochemical measurements showed that most biochemical markers were within normal ranges, except for higher levels of lactate dehydrogenase (LDH, *p* = 0.03) and lower glucose levels (*p* = 0.01) in the patient group. An age disparity was identified among the groups, with the mean age of the AD patient group being significantly higher (*p* = 0.00001). Furthermore, AD patients presented with a possible build-up of intracellular Aβ and extracellular tau, evidenced by reduced cognitive test scores for Mini-Mental State Examination (MMSE, 20.0 ± 4.5) and Addenbrooke’s Cognitive Examination (ACE, 58.0 ± 16.5), and elevated CSF Aβ (682.8 ± 216.3 ng/L), alongside higher Functional Activities Questionnaire (FAQ, 11.8 ± 6.2) cognitive test scores, and increased CSF phospho-tau (p-tau, 81.7 ± 25.0 ng/L) and total-tau (t-tau, 520.4 ± 102.4 ng/L) levels. Plasma measurements of Aβ, tau, glial fibrillary protein (GFAP), and Nf-L by single molecule array (SIMOA) were incorporated to supplement the clinical data. The results indicated significantly raised levels of Aβ_40_ (*p* = 0.002), GFAP (*p* = 0.01), Nf-L (*p* = 0.04), and p-tau181 (*p* = 0.00005) in AD patients, even after adjusting for age-related variations in Nf-L and GFAP (Table 1). Unadjusted values for mean and SD are presented in Table 1.

### 2.2. Cognitive Impairment and Lipoproteins

A comparative analysis of lipoprotein profiles between controls and patients’ unadjusted values revealed that 17 lipoprotein subfractions were significantly dysregulated; however, after age adjustment, only two of these lipoproteins were found to be significantly dysregulated (Table 2). None of the 112 lipoprotein subfractions were found to be significantly altered after false discovery rate (FDR) correction. HDL and LDL subfractions were significantly elevated in the AD group, while the VLDL subclasses were significantly lower. The unadjusted values for mean, standard deviation (SD), and fold change (FC) are presented in Table 2.

The correlations between these significantly altered lipoprotein subfractions and the total lipoprotein levels measured by standard clinical tests were then established (Figure 1). The total measured triglycerides exhibited a strong positive correlation with VLDL subfractions; VLDL free cholesterol (VLFC) (ρ = 0.79), VLDL phospholipids (VLPL) (ρ = 0.77), VLDL-1 triglycerides (V1TG) (ρ = 0.74), VLDL-1 cholesterol (V1CH) (ρ = 0.74), VLDL-1 free cholesterol (V1FC) (ρ = 0.72), and VLDL-1 phospholipids (V1PL) (ρ = 0.74), and similarly for total triglycerides (TPTG) (ρ = 0.78), VLDL triglycerides (VLTG) (ρ = 0.77), and intermediate-density lipoprotein triglycerides (IDTG) (ρ = 0.74).

Subsequently, correlations were established between the lipoprotein subfractions of interest and neurocognitive test scores, levels of AD CSF markers, and plasma measurements of neurodegenerative markers as determined by the SIMOA. The VLDL subfractions VLFC and VLPL exhibited moderate positive and significant correlations with ACE scores (ρ = 0.6 and ρ = 0.6, respectively) and V1PL, with a moderate negative correlation with Aβ_42_ (ρ = −0.54). Conversely, triglyceride containing subfractions TPTG, VLTG, and IDTG demonstrated moderate positive correlations with Aβ_40_ (ρ = 0.54 for TPTG and ρ = 0.54 IDTG) and ACE (ρ = 0.54 for TPTG, ρ = 0.6 for VLTG, and ρ = 0.54 for IDTG) (Figure 2).

Finally, the discriminatory ability of the lipoprotein subfractions in distinguishing between healthy and diseased individuals was examined using receiver operating characteristic (ROC) curve analysis (Figure 3). Among the 17 lipoproteins that showed significant regulation, two exhibited an area under the curve (AUC) of around 0.7, including HDL-1 Apo-A2 (H1A2, AUC = 0.67, 95% CI = 0.52–0.82) and HDL-2 Apo-A2 (H2A2, AUC = 0.67, 95% CI = 0.52–0.82). Despite adjusting for age and sex differences, the significance of these two lipoproteins persisted.

## 3. Discussion

In this study, we compared the levels and composition of various lipoproteins, including their subfractions, to clinical data and diagnostic outcomes to investigate their potential association with cognitive impairment. Moreover, we evaluated the diagnostic capabilities of selected lipoproteins to determine their effectiveness as potential blood-based biomarkers.

A previous study of a similar scope (but reporting no SIMOA-based measurements of neurodegenerative-related proteins in plasma) investigated lipoproteins in AD using NMR with a secondary focus on ApoE status [26]. In brief, certain ApoE genotypes are risk factors for AD, with the Apoε4 allele being a major risk factor and Apoε3 to a lesser extent, whereas Apoε2 plays a protective role in AD [27]. Both studies were in agreement that the HDL subfraction H1A2 is significantly elevated in the AD group compared to healthy controls. As previously stated, HDL has been linked to a neuroprotective role, which is also the case for the ApoA-II subfractions, H1A2 and H2A2. Several apolipoproteins, such as ApoA-II, are not generated in the brain environment; however, HDL lipoproteins have been shown to be able to traverse the BBB [28]. The subfraction ApoA-II is the second most significant constituent of HDL particles [29].

ApoA-II has the capability to form complexes with ApoE2 and ApoE3, resulting in the reduction of internalization into the cell of Aβ by binding to this neurotoxic protein. However, these features are absent with ApoE4 [30]. Of note, ApoA-I and –IV are increased at repair sites of peripheral nerve injuries [31], and a similar mechanism could perhaps apply to ApoA-II. Even though H1A2 and H2A2 correlated moderately with routinely measured HDL cholesterol, no significant difference was observed between the AD and control group for routinely measured HDL, supporting the previous observation that the role of HDL in AD depends on apolipoprotein composition rather than lipoprotein particle quantity. In line with its neuroprotective functions, ApoA-II has been associated with lower CSF-NfL levels in multiple sclerosis [32]. In contrast to these findings, other studies have found lower plasma ApoA-II levels in patients with AD compared to controls in a Japanese cohort [33] and, in mild cognitive impairment (MCI) patients, lower ApoA-II increased the risk of cognitive decline [34]. 

Some discrepancies were also found between the aforementioned study by Berezhony et al. [26] and this present study. Their results highlighted the HDL-4 subfractions Apo-A1 (H4A1), free cholesterol (H4FC), and cholesterol (H4TG) as significantly increased in AD patients, which the present study did not find, possibly due to a smaller cohort size. This was also evident for the LDL and VLDL subfractions LDL-2 cholesterol (L2CH) and VLDL-2 triglycerides (V2TG), which presented similar findings. However, a few lipoproteins, including V1CH, V1TG, and IDTG, were found to be significantly altered in both studies, albeit the alterations were in opposite directions, with these lipoproteins being of lower concentrations in AD patients in the current study. Our findings on VLDL subfractions diverge from previous research, which could be due to possible methodological differences such as NMR protocols or sample preparation, population heterogeneity involving variations in disease stage or genetic backgrounds, and the dynamic nature of lipid metabolism in AD. To resolve these discrepancies, future research should focus on standardizing methods and expanding to larger, more diverse cohorts. Furthermore, these alterations in the current study were not found to be significant post-FDR correction, minimizing their contribution to aiding in differentiating AD patients from healthy individuals.

Given these findings, it is crucial to acknowledge the limitations of this proof-of-concept study. The inclusion of clinical data from patients depended on the physician’s discretion for confirming their diagnosis, resulting in some patients having only neurocognitive test scores available for analysis. In addition, a significant difference was observed in the mean age between the groups, with the AD patient group exhibiting a higher mean age than the healthy control group. Due to age limitations for healthy blood donors, recruiting older individuals for the control group was not feasible. To mitigate this, the lipoprotein values were adjusted for age before conducting the statistical analysis, aiming to eliminate potential effects arising from age disparities. Age-related changes in lipoprotein metabolism are complex and may not be fully accounted for by this adjustment. Although we cannot exclude the possibility that some of the observed differences in lipoprotein profiles between groups could be partially attributed to age rather than AD status alone, available data concerning lipoprotein levels at the ages of the AD patients and healthy controls do not indicate any differences [35]. In addition, our study was limited by the lack of stratification of AD patients by disease stage; however, they were all diagnosed with mild (to moderate) AD at the time of clinical examinations. APOE status by genotyping has also been shown to an important factor in aiding the stratification of the AD group, which our study could benefit from having included, but the study was too small for a stratification based on genotypes. Another limitation to mention is that blood samples were drawn in non-fasting individuals, which could affect their lipoprotein profiles. However, fasting and non-fasting blood samples should be comparable in regards to lipoprotein profiles, with small differences, especially for HDL [36], and the non-fasting state is actually the prevailing condition during the day. Furthermore, small populations were used in this study, thus affecting the ability of the suggested lipoprotein subfractions to differentiate between healthy and diseased individuals. Lastly, it is important to acknowledge that certain confounding factors that could potentially influence the lipoprotein profile, such as smoking, BMI, type 2 diabetes, cardiovascular disease, and dietary supplementation, have not been considered in this study. It should be noted, however, that all the routine blood samples were within reference intervals (including glucose and LDL- and HDL-cholesterol and triglycerides) and not different between the AD and the control groups. Given these limitations, our findings should be interpreted with caution. The observed differences in lipoprotein profiles between AD patients and controls may be partially influenced by these unaccounted factors, rather than solely by AD pathology. Further research is needed to disentangle the effects on lipoprotein profiles of AD from those of other health and lifestyle factors.

Overall, our research improves upon the understanding of AD by highlighting several key aspects: Firstly, it underscores the significant role of lipid metabolism in AD pathology, aligning with the broader recognition of metabolic dysfunction in neurodegenerative disorders. Secondly, our findings support the development of blood-based biomarkers for AD. Lastly, by integrating lipoprotein profiles with established biomarkers such as Aβ and tau, we offer a more comprehensive approach to understanding AD pathology, which may enhance diagnostic precision. These insights pave the way for further exploration into the intricate relationships between lipid metabolism, neuroinflammation, and neurodegeneration in AD.

## 4. Materials and Methods

### 4.1. Characteristics of Study Participants

For this study, 25 patients diagnosed with mild to moderate AD and 25 healthy controls were enrolled. All subjects were Caucasian. Recruitment was performed consecutively at the time of diagnosis for the patients ( ≥65 years) and prior to starting their treatment regimen at the Department of Neurology, Aalborg University Hospital. The patient diagnosis was based on the following criteria: the International Classification of Diseases and Related Health Problems 10th Edition (ICD_10_) [37], and the National Institute of Neurological and Communicative Disorders and Stroke and the Alzheimer’s Disease and Related Disorders Association (NINCDS-ADRDA) [38]. Neurocognitive examinations comprised a MMSE, ACE, and a FAQ. Aβ, p-tau, and t-tau were measured in CSF using Innotest^®^ β-Amyloid(1–42) (Innotest^®^, Triolab, Brøndby, Denmark), Innotest^®^ Phospho-tau (181P) (Innotest^®^, Triolab, Brøndby, Denmark), and Innotest^®^ hTau Ag (Innotest^®^, Triolab, Brøndby, Denmark), respectively, and according to the manufacturer’s instructions. Measurements of CSF and cognitive tests were included when deemed necessary due to diagnostic uncertainty. Exclusion criteria included other forms of dementia or neurological diseases, as well as severe psychiatric disorders. As a control group for comparison with AD patients, age- and sex-related donors were recruited from the blood bank of Aalborg University Hospital. Inclusion criteria for blood donors required them to be ≥65 years old and to complete a standard blood bank questionnaire describing their physical and mental health, such as experiencing memory impairment, fatigue, and chest pain. Exclusion criteria included other forms of dementia or neurological diseases, as well as severe psychiatric disorders. All participants signed a written consent form before inclusion in the study. The study was approved by the local North Denmark Region Committee on Health Research Ethics (N-20150010) and conducted according to the Declaration of Helsinki.

In addition to diagnostic data, routine analyses and markers of neurodegeneration were added to examine possible co-morbidities and characteristics of participants. Briefly, markers of organ function included: alanine transaminase, albumin (Bromocresol Green), cholesterol, creatinine (enzymatic method), C-reactive protein, glucose, HDL, LDL, LDH, triglyceride, and urea, measured using the Alinity c system with dedicated reagents (Abbott, Chicago, IL, USA) [39]. More information on the methods can be obtained from General Chemistry|Core Laboratory at Abbott. Haemoglobin was measured either using the Hb 201 DM (Hemocue AB, Ängelholm, Sweden) or the XN-9000 (Sysmex Europe SE, Norderstedt, Germany) in accordance with the manufacturer’s instructions. Lastly, levels of markers for neurodegeneration included Aβ_40_, Aβ_42_, GFAP, Nf-L, and p-tau181 and were measured by SIMOA^®^ HD-X Analyzer using the commercially available kits, Neurology 4-Plex E and P-Tau181 (Quanterix^©^, Billerica, MA, USA), according to the manufacturer’s instructions. A flowchart of the methods used can be seen in Figure 4.

### 4.2. Collection and Processing of Blood Samples

Collection of blood samples and post-processing were performed as previously described [40]. Briefly, a 21-gauge needle was used to draw blood from the median cubital vein into 10 mL clot activator tubes (BD Vacutainer, UK) and 4 mL ethylenediaminetetraacetic acid (EDTA) tubes. The collected samples were centrifuged twice at 2500× *g* for 15 min at room temperature to acquire serum and plasma samples. After centrifugations, samples were aspirated to approximately 1 cm above the buffy coat or pellet, aliquoted, snap-frozen using liquid nitrogen, and stored at −80 °C until further analyses. EDTA plasma was used for the analysis of markers of neurodegeneration by SIMOA, and serum was used for NMR spectroscopy and routine analysis of biochemical parameters.

### 4.3. Nuclear Magnetic Resonance Spectroscopy

^1^H-NMR spectroscopy was performed as previously described [25]. Briefly, serum samples were thawed for 1 h, carefully diluted at 1:1 dilution with sodium phosphate buffer (0.075 M, pH 7.4, 20% D_2_O in H_2_O, 6 mM NaN3, 4.6 mM 3-(trimethylsilyl)-2,2,3,3-tetradeuteropropanoic acid (TSP-d4)), and aliquoted into 5 mm NMR tubes. Spectra were recorded with a Bruker Avance Neo 600 MHz spectrometer attached to a BBI probe (Bruker Biospin GmbH, Rheinstetten, Germany). An IconNMR (Topspin 4.1.1, Bruker Biospin GmbH, Rheinstetten, Germany) and a Samplejet autosampler (Bruker Biospin GmbH, Rheinstetten, Germany) were used for handling samples and acquisition of data. Using acquisition parameters from Dona et al. [41], one-dimensional nuclear Overhauser effect (1D-NOESY) spectra were recorded at 310 K using a total of 96k data points, 30 ppm spectral width, 32 scans, and water suppression (25 Hz) during relaxation delay (4 s), and mixing time (10 ms). Fourier transformation was applied on free induction decays after artificial zero fillings up to 128k data points and 0.3 Hz line broadening. B.I. Methods (Bruker Biospin GmbH, Rheinstetten, Germany) reference samples were routinely measured and processed in automation for temperature calibration, water suppression determination, and external quantitative referencing, in accordance with the manufacturer’s recommendations. B.I.LISA^TM^ (Bruker Biospin GmbH, Rheinstetten, Germany) was used to automatically quantify lipoprotein subfractions. The subfractions are numbered according to increasing density.

### 4.4. Data Analysis

Extensive lipoprotein profiling was performed using ^1^H-NMR. The analysis revealed distinguished lipoprotein subclasses for cognitively affected patients, based on a total profile of 112 lipoprotein subfractions. Lipoprotein values were corrected for age and sex using generalized additive models. Normal distributions of data and residuals were assessed by the Shapiro–Wilk test. Comparisons between groups were performed using a Mann–Whitney U test with a significance level of *p* < 0.05 for both unadjusted and adjusted values. Data were presented as mean with SD. Correlations between adjusted lipoprotein concentrations and clinical parameters were investigated using Spearman’s ρ. ROC curves were created for adjusted lipoproteins of interest and are presented with an AUC with 95% confidence interval (CI) and *p*-value. R version 4.2.2 was used for data analysis and graphical visualization. The raw NMR lipoprotein data and biochemical data can be accessed in Appendix A and Appendix A, respectively.

## 5. Conclusions

In conclusion, our study makes a significant contribution to the understanding of lipoprotein changes in AD, particularly emphasizing the role of certain lipoprotein subfractions, such as those within the HDL group like H1A2 and H2A2. However, the need for further validation in larger, more diverse patient cohorts is imperative to fully ascertain the diagnostic value of these lipoprotein subfractions. This study establishes a strong foundation for future investigations and accentuates the importance of exploring lipoproteins as potential blood-based biomarkers in the area of neurodegenerative diseases.

## Figures and Tables

**Figure 1 ijms-25-11701-f001:**
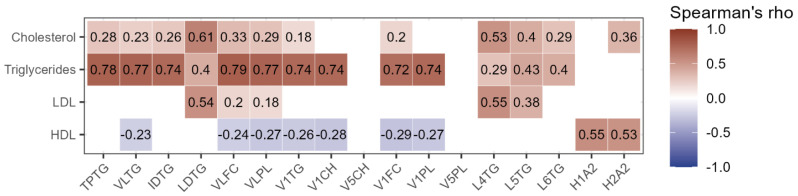
Correlogram of lipoproteins measured by NMR and routine analyses. Only significant correlations are shown. The colour indicates whether the correlation is positive (red) or negative (blue), and the intensity of the colour corresponds to Spearman’s ρ. Abbreviations; HDL—High-density lipoprotein, H1A2—HDL-1 Apo-A2, H2A2—HDL-2 Apo-A2, IDTG—IDL triglycerides, LDL—Low-density lipoprotein, LDTG—LDL triglycerides, L4TG—LDL-4 triglycerides, L5TG—LDL-5 triglycerides, L6TG—LDL-6 triglycerides, TPTG—Total triglycerides, VLFC—VLDL free cholesterol, VLPL—VLDL phospholipids, VLTG—VLDL triglycerides, V1CH—VLDL-1 cholesterol, V1FC—VLDL-1 free cholesterol, V1PL—VLDL-1 phospholipids, V1TG—VLDL-1 triglycerides, V5CH—VLDL-5 cholesterol, and V5PL—VLDL-5 phospholipids.

**Figure 2 ijms-25-11701-f002:**
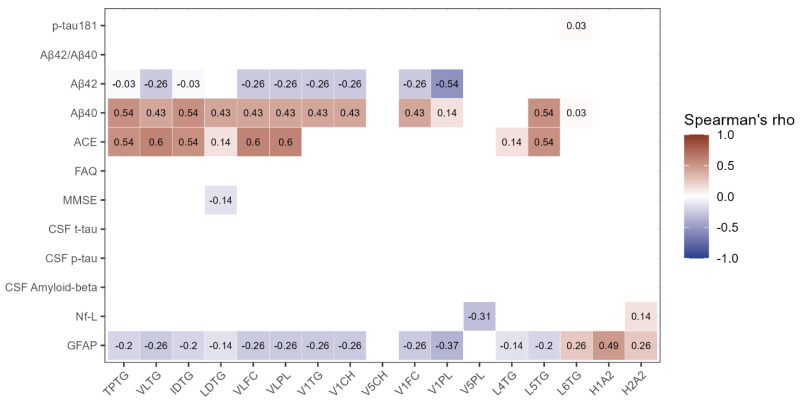
Correlogram of lipoproteins of interest and clinical parameters. Only significant correlations are shown. The colour indicates whether the correlation is positive (red) or negative (blue), and the intensity of the colour corresponds to Spearman’s ρ. Abbreviations; ACE—Addenbrooke’s cognitive examination, FAQ—Functional Activities Questionnaire, GFAP—Glial fibrillary acidic protein, H1A2—HDL-1 Apo-A2, H2A2—HDL-2 Apo-A2, IDTG—IDL triglycerides, LDTG—LDL triglycerides, L4TG—LDL-4 triglycerides, L5TG—LDL-5 triglycerides, L6TG—LDL-6 triglycerides, MMSE—Mini-Mental State examination, Nf-L—Neurofilament light, p-tau—Phospho-tau, t-tau—Total-tau, TPTG—Total triglycerides, VLFC—VLDL free cholesterol, VLPL—VLDL phospholipids, VLTG—VLDL triglycerides, V1CH—VLDL-1 cholesterol, V1FC—VLDL-1 free cholesterol, V1PL—VLDL-1 phospholipids, V1TG—VLDL-1 triglycerides, V5CH—VLDL-5 cholesterol, and V5PL—VLDL-5 phospholipids.

**Figure 3 ijms-25-11701-f003:**
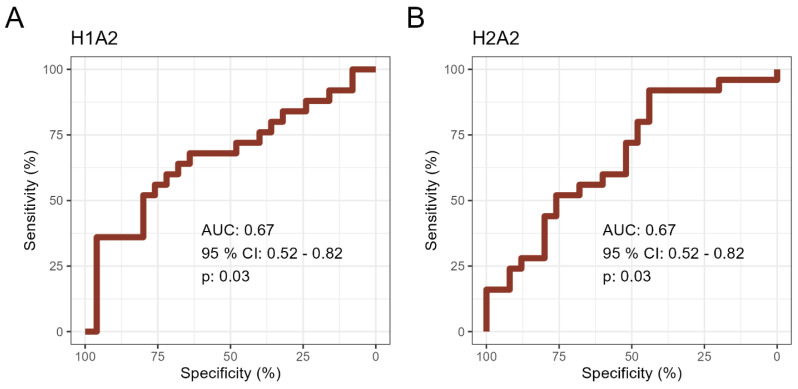
Receiver operating characteristics (ROC) curves for lipoproteins with an area under the curve (AUC) > or ~0.7. ROC curves representing the ability of lipoproteins to differentiate between AD patients and healthy individuals for (**A**) HDL–1 Apo-A2 (H1A2) and (**B**) HDL–2 Apo–A2 (H2A2). AUCs with 95% confidence interval (CI) and *p*–values are reported.

**Figure 4 ijms-25-11701-f004:**
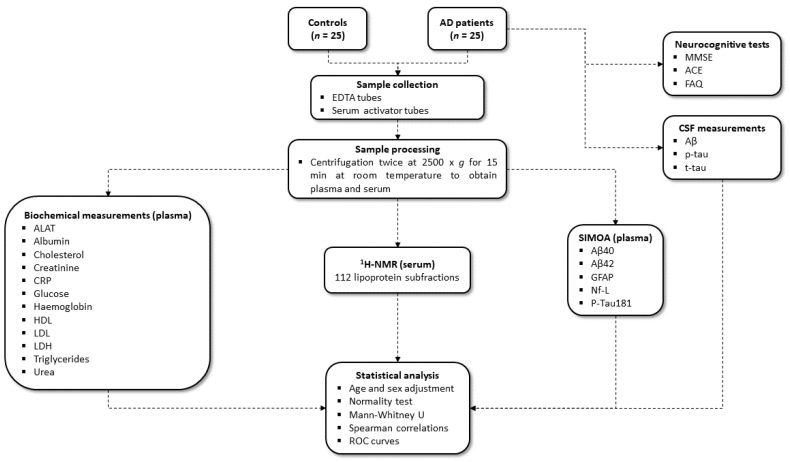
Flowchart of the methods and statistical analysis included. Abbreviations: ^1^H NMR—Proton nuclear magnetic resonance, Aβ—Amyloid-β, ACE—Addenbrooke’s Cognitive Examination, AD—Alzheimer’s disease, ALAT—Alanine transaminase, Con–Healthy control, CRP—C-reactive protein, CSF—Cerebrospinal fluid, EDTA—Ethylenediaminetetraacetic acid, FAQ—Functional Activities Questionnaire, GFAP—Glial fibrillary protein, HDL—High-density lipoprotein, LDH—Lactate dehydrogenase, LDL—Low-density lipoprotein, MMSE—Mini-Mental State Examination, Nf-L—Neurofilament light, p-tau—Phospho-tau, ROC—Receiver operating characteristic, SIMOA—Single molecule array, t-tau—Total-tau.

**Table 1 ijms-25-11701-t001:** Characteristics of study groups. Abbreviations; Aβ—Amyloid-β, ACE—Addenbrooke’s Cognitive Examination, AD—Alzheimer’s disease, ALAT—Alanine transaminase, Con—Healthy control, CRP—C-reactive protein, CSF–Cerebrospinal fluid, FAQ—Functional Activities Questionnaire, GFAP—Glial fibrillary protein, HDL—High-density lipoprotein, LDH—Lactate dehydrogenase, LDL—Low-density lipoprotein, MMSE—Mini-Mental State Examination, Nf-L—Neurofilament light, p-tau—Phospho-tau, SD—Standard deviation, t-tau—Total-tau.

	Con (*n* = 25)	AD (*n* = 25)	*p*-Value	Reference Interval
	Mean (SD)	Mean (SD)		
Demographics				
Age [years]	66.6 (1.3)	75.7 (8.2)	0.00001	-
Male/female [*n*]	16/9	15/10	-	-
Ethnicity	Caucasian	Caucasian	-	-
Biochemical characteristics				
ALAT [U/L]	26.3 (8.6)	22.3 (11.6)	0.17	10.0–50.0
Albumin [g/L]	41.0 (1.9)	41.5 (1.9)	0.37	34–45
Cholesterol [mmol/L]	5.4 (0.9)	5.5 (1.1)	0.88	4.2–8.5
Creatinine [µmol/L]	79.0 (10.2)	83.4 (14.5)	0.22	45–105
CRP [mg/L]	1.9 (1.4)	2.2 (2.9)	0.57	<8
Glucose [mmol/L]	6.4 (1.7)	5.4 (0.9)	0.01	4.2–7.8
Haemoglobin [mmol/L]	8.8 (0.7)	8.5 (1.0, *n* = 15)	0.45	7.3–10.5
HDL [mmol/L]	1.5 (0.3)	1.6 (0.4)	0.35	0.7–1.9
LDL [mmol/L]	3.2 (0.8)	3.3 (0.9)	0.71	2.2–5.7
LDH [U/L]	170.2 (31.2)	192.1 (38.7)	0.03	105–255
Triglycerides [mmol/L]	1.5 (0.8)	1.3 (0.8)	0.34	0.6–3.9
Urea [mmol/L]	5.8 (1.3)	5.7 (1.5)	0.77	3.1–8.1
Neurocognitive test scores				
MMSE	-	20.0 (4.5)	-	-
ACE	-	58.0 (16.5, *n* = 21)	-	-
FAQ	-	11.8 (6.2, *n* = 21)	-	-
CSF neurodegenerative markers				
Aβ [ng/L]	-	682.8 (216.3, *n* = 9)	-	>500
p-tau [ng/L]	-	81.7 (25.0, *n* = 9)	-	<61
t-tau [ng/L]	-	520.4 (102.4, *n* = 9)	-	<450
Plasma neurodegenerative markers				
Aβ_40_ [pg/mL]	95.1 (10.2)	108.7 (17.4)	0.002	-
Aβ_42_ [pg/mL]	5.3 (1.0)	5.6 (1.3)	0.5	-
Aβ_42_/Aβ_40_	0.06 (0.009)	0.05 (0.009)	0.06	-
GFAP [pg/mL]	88.6 (32.8)	247.1 (277.9)	0.01	-
Nf-L [pg/mL]	12.5 (4.4)	36.9 (24.5)	0.04	-
p-tau181 [pg/mL]	1.8 (0.8)	3.1 (1.3)	0.00005	-

**Table 2 ijms-25-11701-t002:** Significantly altered lipoproteins. Both *p*-value and FDR comparisons are shown for unadjusted and adjusted data. Abbreviations; AD—Alzheimer’s disease, Con—Healthy control, FDR—False discovery rate, FC—Fold change, HDL—High-density lipoprotein, H1A2—HDL-1 Apo-A2, H2A2—HDL-2 Apo-A2, IDTG—IDL triglycerides, LDL–Low-density lipoprotein, LDTG—LDL triglycerides, L4TG—LDL-4 triglycerides, L5TG—LDL-5 triglycerides, L6TG—LDL-6 triglycerides, SD—Standard deviation, TPTG—Total triglycerides, VLFC—VLDL free cholesterol, VLPL—VLDL phospholipids, VLTG—VLDL triglycerides, V1CH—VLDL-1 cholesterol, V1FC—VLDL-1 free cholesterol, V1PL—VLDL-1 phospholipids, V1TG—VLDL-1 triglycerides, V5CH—VLDL-5 cholesterol, and V5PL—VLDL-5 phospholipids.

Lipoprotein [g/L]	Con		AD		FC	Unadjusted		Adjusted	
	Mean	SD	Mean	SD		*p*-Value	FDR	*p*-Value	FDR
H1A2	0.030	0.015	0.041	0.018	0.4	0.02	0.2	0.04	1
H2A2	0.037	0.012	0.043	0.012	0.2	0.03	0.3	0.04	1
IDTG	0.173	0.126	0.122	0.135	−0.3	0.02	0.2	0.95	1
L4TG	0.021	0.009	0.030	0.013	0.4	0.01	0.2	0.2	1
L5TG	0.020	0.011	0.027	0.010	0.3	0.03	0.3	0.2	1
L6TG	0.040	0.016	0.049	0.013	0.2	0.01	0.2	0.07	1
LDTG	0.179	0.043	0.226	0.081	0.3	0.02	0.2	0.3	1
TPTG	1.505	0.672	1.258	0.758	−0.2	0.03	0.3	0.95	1
V1CH	0.084	0.050	0.054	0.051	−0.4	0.002	0.2	0.4	1
V1FC	0.036	0.024	0.022	0.026	−0.4	0.006	0.2	0.5	1
V1PL	0.085	0.053	0.060	0.061	−0.3	0.01	0.2	0.6	1
V1TG	0.550	0.321	0.384	0.402	−0.3	0.005	0.2	0.6	1
V5CH	0.014	0.007	0.010	0.006	−0.3	0.05	0.3	0.3	1
V5PL	0.019	0.006	0.014	0.006	−0.3	0.01	0.2	0.2	1
VLFC	0.100	0.036	0.083	0.044	−0.2	0.04	0.3	0.5	1
VLPL	0.231	0.087	0.196	0.112	−0.1	0.04	0.3	0.8	1
VLTG	0.951	0.498	0.768	0.604	−0.2	0.03	0.3	0.9	1

## Data Availability

The original contributions presented in the study are included in the article/Appendix A; further inquiries can be directed to the corresponding author.

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
