# Peer review of "Serum Lipoprotein Profiling by NMR Spectroscopy Reveals Alterations in HDL-1 and HDL-2 Apo-A2 Subfractions in Alzheimer’s Disease"

_ijms, 2024, doi:10.3390/ijms252111701_

Round 1
Reviewer 1 Report
Comments and Suggestions for Authors
Blood lipoprotein profiling has emerged as a promising method for identifying biomarkers for AD. This profiling assesses the levels and types of lipoproteins in the blood, which are associated with lipid metabolism and amyloid-beta (Aβ) clearance—both crucial aspects of AD pathogenesis. This manuscript supplied a new viewpoint of the lipoprotein subfractions in healthy control and AD patients. There are several major and minor weaknesses in the rationale and research methods of this work. Below please find the review comments.
(1) Major comments
1. Rewrite the title. The author needs to revise the title to make it match the statement within this manuscript. For example, " Changes in Hdl Subfractions " is not a standard term in neuroscience or medical literature, it could be associated with any components that constitute lipoprotein profiles, so it is better to refer a more specific concept or scientific context.
2. It is better for the author to supply a workflow chart after the method section, so that the authors can read the procedure, methods, results, and findings within this manuscript in the easy way.
3. Please be cautious to use the “biomarker” at the end of abstract. Biomarkers are biological indicators that can be measured to detect or monitor a disease state, predict disease risk, or evaluate the effectiveness of treatments. They are essential tools in clinical practice and research for diagnosing, prognosing, and tracking the progress of diseases, as well as determining responses to therapies. Here in this manuscript, I do not think the data is sufficient to support this point.
4. Based on my knowledge, AD is a heterogenous disease, and the stage is closely correlated with recognitive deficient, I did not see this point in the AD groups. Moreover, the BMI in control and AD groups is also a pivotal factor for the lipoprotein subfractions in blood, it needs more data and evidence to support this point, i.e., lipoprotein subfractions have potential as biomarkers for AD diagnosis.
5. Page 2, line 72, I think ref 17 is using animal models to indicate the accumulation of lipid droplets in AD pathology, rather than human patients, it is essential to address it.
6. Page 3, line 110, Table 1, it is essential to include clear inclusion and exclusion criteria for patient recruitment. Additionally, comprehensive clinical data, including factors such as BMI, APOE types, and potential comorbidities commonly seen in aging populations, should be accounted for. This will ensure the analysis is not biased by other underlying conditions that may affect the biochemical analysis and lipoprotein outcomes.
(2) Minor comments
1. Abbreviations should be properly defined and interpreted when they are first introduced in the manuscript. This practice ensures that readers can understand the meaning of abbreviations without confusion. It is also recommended to include a comprehensive list of abbreviations after the conclusion section. This list will serve as a quick reference for readers, enabling them to easily access the definitions throughout the manuscript.
Author Response
Question 1. Rewrite the title. The author needs to revise the title to make it match the statement within this manuscript. For example, " Changes in Hdl Subfractions " is not a standard term in neuroscience or medical literature, it could be associated with any components that constitute lipoprotein profiles, so it is better to refer a more specific concept or scientific context.
Response 1: We agree that the title should be more specific. The title has been revised to :"Serum Lipoprotein Profiling by NMR Spectroscopy Reveals Alterations in HDL-1 and HDL-2 Apo-A2 Subfractions in Alzheimer's Disease", page 1, lines 2 - 4.
Question 2. It is better for the author to supply a workflow chart after the method section, so that the authors can read the procedure, methods, results, and findings within this manuscript in the easy way.
Response 2. We agree that adding a workflow chart after the methods section would improve readability. We have created a flowchart to include (page 10, lines 363 – 371):
- Patient recruitment and grouping
- Sample collection and processing
- NMR spectroscopy analysis
- Data analysis and statistical methods
Question 3. Please be cautious to use the “biomarker” at the end of abstract. Biomarkers are biological indicators that can be measured to detect or monitor a disease state, predict disease risk, or evaluate the effectiveness of treatments. They are essential tools in clinical practice and research for diagnosing, prognosing, and tracking the progress of diseases, as well as determining responses to therapies. Here in this manuscript, I do not think the data is sufficient to support this point.
Response 3. Thank you for bringing this important point to our attention. We agree that the use of the term "biomarker" in the abstract's conclusion should be more cautious, given the current data. We propose the following revision to address this concern:
Revision: "These results indicate that these lipoprotein subfractions may have potential as indicators of AD-related metabolic changes”. Page 1, lines 33 – 34.
This revision:
- Removes the direct claim of these subfractions being biomarkers.
- Acknowledges their potential as indicators of AD-related changes.
We believe this more conservative statement better reflects the current state of our findings while still highlighting the potential significance of these lipoprotein subfractions in AD research.
Question 4. Based on my knowledge, AD is a heterogenous disease, and the stage is closely correlated with recognitive deficient, I did not see this point in the AD groups. Moreover, the BMI in control and AD groups is also a pivotal factor for the lipoprotein subfractions in blood, it needs more data and evidence to support this point, i.e., lipoprotein subfractions have potential as biomarkers for AD diagnosis.
Response 4. Thank you for this important observation. You are correct that our study has limitations regarding the heterogeneity of AD and the potential influence of BMI on lipoprotein profiles. We acknowledge these points and propose the following response:
- AD heterogeneity: We recognize that we did not stratify AD patients by disease stage or cognitive deficit severity. This may be a limitation of our study. However, all the patients were included at the time where they were admitted to be examined for the presence of AD, i.e. the patients had mild (to moderate) AD. Since it is the time for diagnosis it is a relevant time to investigate potential changes, which could help in diagnostics. Patients with mild (to moderate) AD were grouped together to encompass patients early in the disease stage to aid in stratifying early disease related changes from healthy individuals.
- BMI consideration: We acknowledge that BMI is indeed a crucial factor influencing lipoprotein subfractions. Unfortunately, BMI data was not collected in this study, which is a limitation. However, we investigated all the lipoproteins, and a difference in BMI between the cases and the controls would mainly appear as differences in LDL and triglyceride fractions, and they were very much alike in the two groups (Table 1).
To address these limitations, we propose adding the following paragraph to our discussion section under limitation: "In addition, our study was limited by the lack of stratification of AD patients by disease stage; however, they were all diagnosed with mild (to moderate) AD at the time of clinical examinations.”, page 7, lines 250 – 252. For BMI data we have addressed this limitation (pages 7 – 8, lines 261 – 271) together with other confounding factors, which could influence the lipoprotein profile.
Question 5. Page 2, line 72, I think ref 17 is using animal models to indicate the accumulation of lipid droplets in AD pathology, rather than human patients, it is essential to address it.
Response 5 . Thank you for bringing this important point to our attention. You are correct that reference 17 describes findings from animal models rather than human patients. We appreciate the need for clarity in distinguishing between animal and human studies.
We propose the following revision to address this concern:
Revised text: "Additionally, studies in animal models have observed that the accumulation of lipid droplets in the AD brain occurs prior to amyloid aggregation, highlighting the potential significance of dysregulated lipoprotein metabolism in neurodegenerative disorders.", page 2, lines 71 – 73.
This revision:
- Clearly states that the observation comes from animal model studies.
- Maintains the importance of the finding while acknowledging its source.
- Uses more cautious language ("potential significance") to reflect that the observation is from animal models and may not directly translate to human AD pathology.
We believe this change provides a more accurate representation of the research findings and helps readers distinguish between animal and human studies in our discussion of AD pathology.
Question 6. Page 3, line 110, Table 1, it is essential to include clear inclusion and exclusion criteria for patient recruitment. Additionally, comprehensive clinical data, including factors such as BMI, APOE types, and potential comorbidities commonly seen in aging populations, should be accounted for. This will ensure the analysis is not biased by other underlying conditions that may affect the biochemical analysis and lipoprotein outcomes.
Response 6. Thank you for this important feedback. We acknowledge that our study would indeed benefit from more comprehensive inclusion and exclusion criteria, as well as additional clinical data.
Inclusion criteria for AD patients:
- Diagnosis of mild to moderate AD based on ICD10 and NINCDS-ADRDA criteria
- Age ≥ 65 years
- Initial diagnosis
Exclusion criteria for AD patients:
- Other forms of dementia or neurological disorders
- Severe psychiatric disorders
- Initiation of treatment regimen for AD
Inclusion criteria for controls:
- Age ≥ 65 years
- No self-reported cognitive impairment
Exclusion criteria for controls:
- Any neurological or psychiatric disorders
This has also been further elaborated on page 8, lines 283 – 302.
Additional clinical data:
We acknowledge the lack of comprehensive clinical data in our current study. We propose to address this limitation in future research by collecting and analyzing the following:
- BMI: We recognize BMI as a crucial factor influencing lipoprotein profiles.
- APOE genotype: We agree that APOE status is important in AD research. However, we do not think that it is very important in this study. In the AD group there will probably be a higher frequency of ε4 but there will be several different genotypes, and our group of patients is not big enough for a subdivision in several genotypes. We wanted to investigate differences in patients diagnosed with AD compared with healthy controls.
- Comorbidities: We could have collected more detailed information on comorbidities. However, we want to stress that all the routine samples of the patients were normal including e.g. glucose and lipids (Table 1).
Limitations section:
We will add the following paragraph to our Discussion section to address these limitations:
"APOE status by genotyping has also been shown to an important factor in aiding the stratification of the AD group, for which our study could benefit from having included, but the study was too small for a stratification based on genotypes.", page 7, lines 252 – 255.
As answered in Question 4, we have addressed the issues regarding BMI and comorbidities such diabetes type 2 and cardiovascular disease (page 7, lines 261 - 271).
(2) Minor comments
Question 7. Abbreviations should be properly defined and interpreted when they are first introduced in the manuscript. This practice ensures that readers can understand the meaning of abbreviations without confusion. It is also recommended to include a comprehensive list of abbreviations after the conclusion section. This list will serve as a quick reference for readers, enabling them to easily access the definitions throughout the manuscript.
Response 7. Thank you for bringing this to our attention, the manuscript has been revised for correct use of abbreviations. We agree that including an abbreviation list will improve the readability of the manuscript and have included it after the conclusions segment, pages 10 – 11, lines 380 – 399.
Reviewer 2 Report
Comments and Suggestions for Authors
The article entitled "Serum lipoprotein profiling by NMR spectroscopy identifies changes in HDL subfractions in Alzheimer's disease” aims to investigate the lipoproteins in the blood of Alzheimer's patients for their diagnostic potential.
There was a significant age difference between the control and AD groups. Could this difference have influenced the results, even though it was adjusted for age? Could older age, independent of AD, affect lipoprotein levels?
The blood samples were taken from people who were not fasting. Could the absence of fasting affect lipoprotein profiles, particularly triglycerides, and how does this affect the reliability of the results?
Important factors such as smoking, BMI, cardiovascular disease and dietary supplements were not taken into account. How might these potential confounders have affected the results of the study?
Some of the results, particularly in relation to lipoprotein subfractions, differ from previous studies. Could methodological differences (e.g. sample preparation, NMR protocols) explain the discrepancies?
The AUC values for HDL subfractions as potential biomarkers were around 0.67, indicating a moderate discrimination between AD patients and healthy controls. Could this value indicate that lipoprotein subfractions are not very specific or sensitive as biomarkers?
The study emphasizes the role of apoA-II in the HDL subfractions. However, could other lipoproteins or metabolic pathways play an equally significant or greater role in Alzheimer’s pathology that have not been investigated?
Author Response
The article entitled "Serum lipoprotein profiling by NMR spectroscopy identifies changes in HDL subfractions in Alzheimer's disease” aims to investigate the lipoproteins in the blood of Alzheimer's patients for their diagnostic potential.
Question 1. There was a significant age difference between the control and AD groups. Could this difference have influenced the results, even though it was adjusted for age? Could older age, independent of AD, affect lipoprotein levels?
Response 1. Thank you for raising this important point about the age difference between the control and AD groups. You are correct that this is a significant limitation of our study that warrants careful consideration.
We acknowledge that the significant age difference between our control group (mean age 66.6 years) and AD group (mean age 75.7 years) is a limitation of this study. While we did adjust for age in our statistical analysis, we recognize that this adjustment may not fully account for all age-related effects on lipoprotein profiles. However, we have no reason to believe that this difference will have a substantial effect, since differences in lipoproteins in these age groups are really minor (see e.g. Hughes D. Irish J Med Sci 2021;190:117). Unfortunately, it is very difficult to recruite elderly persons for controls (donors have an age limit).
Given these age-related changes, we propose the following additions to our limitations section:
"In addition, a significant difference was observed in the mean age between the groups, with the AD patient group exhibiting a higher mean age than the healthy control group. Due to age limitations for healthy blood donors, recruiting older individuals for the control group was not feasible. To mitigate this, the lipoprotein values were adjusted for age before conducting the statistical analysis, aiming to eliminate potential effects arising from age disparities. Age-related changes in lipoprotein metabolism are complex and may not be fully accounted for by this adjustment. Although we cannot exclude that some of the observed differences in lipoprotein profiles between groups could be partially attributed to age rather than AD status alone, available data concerning lipoprotein levels at the ages of the AD patients and healthy controls do not indicate any differences [35].", page 7, lines 240 – 250.
Question 2. The blood samples were taken from people who were not fasting. Could the absence of fasting affect lipoprotein profiles, particularly triglycerides, and how does this affect the reliability of the results?
Response 2. As indicated in reference 36 in our manuscript, that lipid profiles do not seem to change significantly between fasting and non-fasting individuals based on the Copenhagen General Population Study. Regarding plasma triglycerides a maximal mean increase of 0.3 mmol/L was found after 3-4 hours postprandial, with decreasing values afterwards, and cholesterol do not change (including HDL-cholesterol). However, we acknowledge this concern raised by the reviewer, but we have no reason to think that it has influenced the results. Furthermore, the non-fasting condition is the condition, which prevails during the day (i.e. potentially a more reliable condition).
Therefore, we have included the following sentence in our limitation:
“Given these limitations, our findings should be interpreted with caution. The observed differences in lipoprotein profiles between AD patients and controls may be partially influenced by these unaccounted factors, rather than solely by AD pathology. Further research is needed to disentangle the effects of AD from those of other health and lifestyle factors on lipoprotein profiles.”, pages 7 – 8, lines 267 – 271.
Question 3. Important factors such as smoking, BMI, cardiovascular disease and dietary supplements were not taken into account. How might these potential confounders have affected the results of the study?
Response 3. Thank you for raising this important point about potential confounding factors. You are correct that our study did not account for several important variables that could influence lipoprotein profiles.
We acknowledge that our study has limitations regarding potential confounding factors that were not accounted for, including smoking status, BMI, cardiovascular disease, and dietary supplements. These factors can indeed influence lipoprotein profiles and potentially affect our results. We have addressed these limitations in the discussion (page 7, lines 261 - 271).
Such confounding factors could affect our results as for example:
- Smoking is known to lower HDL cholesterol levels and may alter the functionality of HDL particles.
- BMI is associated with lipoprotein profiles. However, we investigated all the lipoproteins, and a difference between the cases and the controls would mainly appear as differences in LDL and triglyceride fractions, and they were very much alike in the two groups (Table 1).
- Cardiovascular disease can alter lipoprotein metabolism and composition.
- Dietary supplements, especially those containing omega-3 fatty acids or plant sterols, can influence lipoprotein levels and composition.
We acknowledge these limitations, but, on the other hand, this study is a “real-world” study where we investigate the patients as they come. The group is not so large that we could make a thorough subdivision in many groups.
We have added the following sentence to our limitations segment:
“Given these limitations, our findings should be interpreted with caution. The observed differences in lipoprotein profiles between AD patients and controls may be partially influenced by these unaccounted factors, rather than solely by AD pathology. Further research is needed to disentangle the effects of AD from those of other health and lifestyle factors on lipoprotein profiles.”, pages 7 – 8, lines 267 – 271.
Question 4. Some of the results, particularly in relation to lipoprotein subfractions, differ from previous studies. Could methodological differences (e.g. sample preparation, NMR protocols) explain the discrepancies?
Response 4. Thank you for highlighting this important point about the discrepancies between our results and previous studies, particularly regarding lipoprotein subfractions. You are correct that methodological differences could potentially explain these discrepancies. We propose the following response to address this concern:
- Methodological difference
- Sample preparation: Our study used a specific protocol for sample preparation, including a 1:1 dilution with sodium phosphate buffer. Variations in sample preparation methods across studies could affect lipoprotein measurements.
NMR protocols: We used a Bruker Avance Neo 600 MHz spectrometer with specific acquisition parameters. Different NMR instruments, field strengths, or acquisition parameters used in other studies could lead to variations in lipoprotein quantification.
Data processing and analysis: Our use of B.I.LISA™ for automatic quantification of lipoprotein subfractions may differ from methods used in other studies, potentially leading to discrepancies in results.
- Population heterogeneity
Our study population may differ from those in previous studies in terms of factors such as:
a. Disease stage: The severity of AD in our patient group may not be directly comparable to other studies.
- Genetic background: Variations in genetic factors influencing lipoprotein metabolism could contribute to differences in results across studies.
- Lifestyle and environmental factors: Differences in diet, physical activity, and other environmental factors between study populations could impact lipoprotein profiles.
- Dynamic nature of lipid metabolism in AD
The progression of AD and its effects on lipid metabolism may not be linear or consistent across all patients or stages of the disease. This could lead to apparent discrepancies between studies depending on the specific characteristics of the study population.
Question 5. The AUC values for HDL subfractions as potential biomarkers were around 0.67, indicating a moderate discrimination between AD patients and healthy controls. Could this value indicate that lipoprotein subfractions are not very specific or sensitive as biomarkers?
Response 5. Thank you for your insightful comments on the AUC values for HDL subfractions H1A2 and H2A2, which indicate moderate discrimination between Alzheimer’s disease patients and healthy controls. These values underscore the limited specificity and sensitivity of these subfractions as standalone biomarkers, which, on the other hand, may indicate a potential important pathophysiological difference.
Acknowledging this, we suggest their potential utility within a broader panel of biomarkers, which could enhance overall diagnostic precision. Comparative analysis with other blood-based AD biomarkers might reveal complementary benefits, underscoring the value of further research to:
- Evaluate combinations with other biomarkers for improved accuracy,
- Investigate predictive value in specific AD subgroups,
- Conduct longitudinal studies to determine their role in tracking disease progression.
Also, it is worth mentioning that the smaller population in this study could affect the performance of the ROC curves, therefore resulting in a smaller AUC. We propose adding a paragraph to the discussion that addresses this consideration:
“Furthermore, small study populations were used in this study, thus affecting the performance of suggested lipoprotein subfractions to differentiate between healthy and diseased individuals.". Page 7, lines 259 – 261.
Question 6. The study emphasizes the role of apoA-II in the HDL subfractions. However, could other lipoproteins or metabolic pathways play an equally significant or greater role in Alzheimer’s pathology that have not been investigated?
Response 6. Thank you for raising an important question about the broader context of lipoproteins and metabolic pathways in Alzheimer's disease (AD) pathology. While our study specifically focused on HDL subfractions containing apoA-II, there are indeed numerous other factors that could significantly influence AD. To elaborate:
Additional Lipoproteins to Consider:
- ApoE: Known for its strong association with AD risk, especially the ApoE4 isoform. Our study did not examine ApoE genotypes or levels, which could be seen as a limitation.
- ApoJ (Clusterin): This lipoprotein has been implicated in amyloid-β clearance and AD pathology.
- ApoB: Recent research suggests a role for ApoB in early tau pathology in AD.
Relevant Metabolic Pathways:
- Insulin signaling: There is growing evidence linking insulin resistance in the brain to AD pathology.
- Mitochondrial dysfunction: Changes in energy metabolism are critical in the context of AD.
- Broader lipid metabolism: Alterations in membrane lipid composition and lipid rafts may also play a role in AD pathology.
Acknowledging Our Study’s Limitations:
- Our analysis was restricted to specific lipoprotein subfractions identifiable by our NMR method.
- The relatively small sample size may have constrained our ability to observe broader changes in lipoproteins or other metabolites.
- The study did not cover intracellular lipid metabolism or brain-specific lipoprotein production, which could be pivotal.
Your inquiry underscores the necessity for a multifaceted approach in future AD research. While our findings on apoA-II-containing HDL subfractions add valuable knowledge, they indeed represent just one component of the complex metabolic landscape involved in AD pathology.
Round 2
Reviewer 1 Report
Comments and Suggestions for Authors
I appreciate the author's responses to my queries and the additional data they have done to address these.
Reviewer 2 Report
Comments and Suggestions for Authors
The authors have responded positively to my concerns.